# Effects of Intermittent and Chronic Hypoxia on Fish Size and Nutrient Metabolism in Tiger Puffer (*Takifugu rubripes*)

**DOI:** 10.3390/ani14172470

**Published:** 2024-08-25

**Authors:** Qiang Ma, Renxiao Zhang, Yuliang Wei, Mengqing Liang, Houguo Xu

**Affiliations:** 1State Key Laboratory of Mariculture Biobreeding and Sustainable Goods, Yellow Sea Fisheries Research Institute, Chinese Academy of Fishery Sciences, Qingdao 266071, China; maqiang@ysfri.ac.cn (Q.M.); renxiaoz2022@163.com (R.Z.); weiyl@ysfri.ac.cn (Y.W.); liangmq@ysfri.ac.cn (M.L.); 2Laboratory for Marine Fisheries Science and Food Production Processes, Qingdao Marine Science and Technology Center, Qingdao 266237, China; 3Key Laboratory of Aquaculture Nutrition and Feed, Ministry of Agriculture, Ocean University of China, 5 Yushan Road, Qingdao 266003, China

**Keywords:** intermittent hypoxia, chronic hypoxia, nutrients metabolism, hypoxia-inducible factor, marine deoxygenation

## Abstract

**Simple Summary:**

Intermittent and chronic hypoxia are common and harmful to marine animals. The tiger puffer is a representative species of the family Tetraodontidae, which has only slit-like gill openings and is susceptible to hypoxia stress. Both intermittent and chronic hypoxia decrease the growth and visceral weight of tiger puffer but increase the feed conversion ratio and blood hemoglobin content. Chronic hypoxia but not intermittent hypoxia promoted protein synthesis and the glycolysis pathway. Intermittent hypoxia but not chronic hypoxia decreased lipid synthesis and monounsaturated fatty acids content but increased n-3 polyunsaturated fatty acids levels. These changes promoted the adaption of tiger puffer to intermittent and chronic hypoxia.

**Abstract:**

Intermittent and chronic hypoxia are common stresses to marine fish, but the different responses of fish to intermittent and chronic hypoxia have not been well-known. In this study, tiger puffers were farmed in normoxia conditions (NO, 6.5 ± 0.5 mg/L), intermittent hypoxia (IH, 6.5 ± 0.5 mg/L in the day and 3.5 ± 0.5 mg/L in the night), or choric hypoxia (CH, 3.5 ± 0.5 mg/L) conditions for 4 weeks, after which the growth, nutrient metabolism and three *hifα* isoforms expression were measured. Both intermittent and chronic hypoxia decreased the fish growth and visceral weight but increased the feed conversion ratio and blood hemoglobin content. Chronic hypoxia but not intermittent hypoxia promoted protein synthesis and whole-fish protein content by activating *mtor* gene expression and promoted the glycolysis pathway by activating gene expression of *hif1α* and *hif2α*. Intermittent hypoxia but not chronic hypoxia decreased the hepatic lipid synthesis by inhibiting *fasn* and *srebf1* gene expression. Meanwhile, intermittent hypoxia reduced the monounsaturated fatty acid content but increased the n-3 polyunsaturated fatty acids percentage. The results of this study clarified the adaptive mechanism of tiger puffer to intermittent and chronic hypoxia, which provides important information about mechanisms of hypoxia adaption in fish.

## 1. Introduction

Dissolved oxygen (DO), expressed as mg/L or saturation (%), is the molecular oxygen (O_2_) dissolved in the water environment, which plays an important role in the survival of aquatic organisms and the homeostasis of water ecosystems [1,2]. Hypoxia stress of aquatic animals usually refers to the DO level < 2 mg/L [3], but some fish species, such as turbot (*Scophthalmus maximus*) and rainbow trout (*Oncorhynchus mykiss*), are intolerant to hypoxia and have severe stress responses when the DO level falls below 4 mg/L. The O_2_ content in the air is stable at about 21%, but the DO level of the natural water environment is affected by water quality and always fluctuates. Therefore, unlike terrestrial animals, aquatic animals are more susceptible to hypoxia stress [4]. According to the duration of hypoxia, hypoxia can be divided into three major types: acute hypoxia, intermittent hypoxia, and chronic hypoxia. In aquaculture, factors such as high temperature, extreme weather, power failure, and long-distance transportation can all reduce the DO level of the water environment, resulting in acute hypoxia stress for aquatic animals. The high-density aquaculture mode of Atlantic salmon needs high DO levels in Norway and Chile, and uncontrollable ocean currents caused acute hypoxia, leading to massive deaths of fish and severe economic losses [5]. In most natural waters, DO level varies with photoperiod, with high DO during the day and low DO at night, resulting in intermittent hypoxia in aquatic animals [6,7]. During the daytime, aquatic plants and algae can perform photosynthesis and produce O_2_ continuously, and the water DO level reaches its highest level before sunset; while at night, aquatic plants, animals, and microorganisms perform respiration, and the water DO content is continuously decreasing, and generally reaches its lowest level before sunrise. Therefore, intermittent hypoxia is a common phenomenon for aquatic animals [8]. Ocean warming, eutrophication, and high-density aquaculture all keep the water DO at low levels for a long time and cause chronic hypoxia, resulting in death or growth inhibition of aquatic animals [9]. With the excessive discharge of greenhouse gases (CO_2_) and wastewater containing nitrogen and phosphorus, the DO content of the Gulf of Mexico and Amazon rivers has been declining, leading to a dramatic decrease in aquatic animal populations and serious damage to the ecosystem [3,10,11]. Therefore, acute, intermittent, and chronic hypoxia are three common types of hypoxia stress in both natural water environments and aquaculture activities.

During oxidative phosphorylation, O_2_ receives electrons from FADH_2_ and NADH of the respiratory chain to produce adenosine triphosphate (ATP) [12]. In hypoxia conditions, the efficiency of electron transport in the respiratory chain would decrease, which would suppress the synthesis of ATP and cause incomplete reduction of O_2_ to generate the reactive oxygen species (ROS) that are harmful to the organism [13,14]. In mammals and fish, the hypoxia-inducible factor (HIF) signaling pathway plays key roles in cellular response to hypoxia stress. The lack of oxygen inhibits the breakdown of HIFα (including HIF1α, HIF2α/EPAS1, and HIF3α/IPAS), and accumulated HIFα could enter into the cell nucleus and bind with the HIFβ. Then, the dimer would combine with the hypoxia-responsive element (HRE) and promote the transcription of target genes [15,16]. Studies in mammals have shown that HIF1α and HIF2α played a major role in hypoxia, activating numerous downstream target genes, such as key genes of glycolysis (GLUT1, HK, and LDH, etc.), vascular endothelial growth factor (VEGF), and erythropoietin (EPO) [17,18]. However, some HIF3α transcripts in mammals are negative regulators of HIF1α and HIF2α by competitively binding to HIFβ to repress the target genes expression under hypoxia [19]. Unlike mammalian HIF3α, zebrafish Hif3α is also an oxygen-dependent transcriptional activator and plays a similar role as Hif1α under hypoxia [20,21]. Oscar (*Astronotus ocellatus*) is a hypoxia-tolerant fish in the Amazon River, and acute hypoxia (DO = 0.5 mg/L for 3 h) increased the red blood cells, hematocrit, hemoglobin, and glucose levels, as well as genes expression of *hif1α* and *vegf* in the liver [22]. Similar results were found in Nile tilapia (*Oreochromis niloticus*) [23,24]. In largemouth bass (*Micropterus salmoides*), acute hypoxia (DO = 1.2–3 mg/L for 4–8 h) significantly increased the serum glucose content, liver contents of lactic acid, triglycerides, and non-esterified fatty acids, as well as the expression of *hif1α*, *glut1*, and glycolysis-related genes [25,26]. These studies all proved that acute hypoxia activated *hif1α* and promoted glucose catabolism by the anaerobic glycolysis pathway in fish. However, the differences in nutrient metabolism patterns and expression of three *hifα* isoforms between intermittent hypoxia and chronic hypoxia are unknown in marine fish species to date.

Tiger puffer (*Takifugu rubripes*) is a representative species of the family Tetraodontidae and is distributed in the northwestern Pacific [27]. Tetraodontids are characterized by a loose abdomen that can be inflated with air or water, a beak-like dental plate divided by a median suture, and tetrodotoxin and saxitoxin in the tissues [28]. The family Tetraodontidae also have the smallest vertebrate genomes yet measured. Tiger puffers have become a model organism for genomics studies, and Japan has approved the sale of CRISPR-edited tiger puffers [29,30]. Tiger puffers are one of the most valuable commercial fish species because of their high nutritional value and delicious flavor. However, the capture production of wild tiger puffers has greatly decreased since the 1980s because of the higher fishing pressure and deteriorating marine environment [31]. Therefore, tiger puffers are widely farmed in China, Japan, and Korea in both floating cages and industrial recirculating aquaculture systems with high density. China is the largest tiger puffer and obscure puffer (*Takifugu obscurus*) producer in the world, and the production was 16,612 and 14,434 tons in 2022, respectively [32]. The tiger puffer is a demersal species with only a slit-like gill opening anterior to the base of the pectoral fin, making them susceptible to hypoxia stress. Therefore, in this study, tiger puffers were reared in normoxia, intermittent, and chronic hypoxia environments for 4 weeks, and then blood, liver, and muscle were collected for indicators measurement. The results of the study illustrated the effects of intermittent and chronic hypoxia on growth and nutrient metabolism in tiger puffers for the first time, as well as improved our understanding of the adaptive mechanisms of different hypoxia patterns in fish.

## 2. Materials and Methods

### 2.1. Fish Culture and Hypoxia Conditions Management

Juvenile tiger puffers (three-months-old) were purchased from Huanghai Aquaculture Co. Ltd. (Yantai, China) and were cultured in a flow-through seawater system. A total of 180 experimental tiger puffers with an average initial body weight of appr. 37 g were distributed into three groups: normoxia group (NO), intermittent hypoxia group (IH), and choric hypoxia group (CH). Each group had triplicate polyethylene tanks (300 L), and each tank had 20 fish. The water dissolved oxygen level of the NO group was kept at 6.5 ± 0.5 mg/L, and the DO level in the CH group was kept at 3.5 ± 0.5 mg/L. In the IH group, the DO level was kept at 6.5 ± 0.5 mg/L during the daytime (from 8:00 to 20:00), but the DO level was kept at 3.5 ± 0.5 mg/L in the night (from 20:00 to 8:00). An air pump was used to control the DO level. All fish were fed the experimental diets following the standard procedures. The formula and proximate compositions of the diets are listed in Table 1. All fish were fed twice per day at 7:00 and 19:00, and the feeding rate was 3–4% of body weight per tank. The water salinity, pH, temperature, and total ammonia nitrogen were maintained at 26.5 ± 2.5‰, 7.5 ± 0.5, 22.5 ± 1 °C, and <0.02 mg/L, respectively.

### 2.2. Fish Sampling and Indicators Calculating

After four weeks of farming in normoxia, intermittent, or chronic hypoxia conditions, all fish were fasted for 12 h, then weighed and counted. After anesthesia with eugenol, nine fish in each group (three fish in each tank) were selected for the collection of blood, liver, and muscle samples. The blood was kept at 4 °C for 2 h, then centrifuged at 3000 rpm for 10 min for serum collection. The serum, liver, and muscle samples were frozen with liquid nitrogen immediately and stored at −80 °C for further analysis. Six fish in each group (two fish in each tank) were selected for the measurement of body weight and length, the weight of the liver and viscera, based on which the hepatosomatic index (HSI), viscerosomatic index (VSI), and condition factor (K) were calculated. Six fish per group (two fish per tank) were used for the analysis of the proximate composition of the whole fish body.

### 2.3. Proximate Analysis of Diet and Whole Fish Compositions

Feed and whole-fish proximate compositions were assayed using the standard methods of AOAC. Weighed feed and whole-fish samples were dried at 105 °C for 24 h to calculate dry matter and moisture levels. The crude protein content was assayed using the Kjeldahl method (N × 6.25). The crude lipid content was detected by the Soxtec 2050 Soxhlet extractor with petroleum ether (Foss, Hilleroed, Denmark). Samples were weighed and put into a muffle furnace at 550 °C for 4 h to calculate the ash content.

### 2.4. Biochemical Indexes Assays

The contents of glucose (F006-1-1), pyruvate (A081-1-1), lactate (A019-2-1), glycogen (A043-1-1), total soluble protein (A045-2-2), and triglyceride (A110-1-1) in the serum, liver, and muscle were determined using commercial kits. All measurement steps refer to relevant kit protocols at http://www.njjcbio.com/ (accessed on 1 August 2024), and the absorbance was read using a microplate reader (Tecan infinite M200, Männedorf, Switzerland). The blood hemoglobin level was assayed using a portable hemoglobin monitor (electrochemistry method, Taiwan BeneCheck, Taipei City, China).

### 2.5. Extraction of RNA and qPCR

The total RNA of the muscle and liver was isolated using an RNAiso Plus kit (Takara, Kyoto, Japan), and the quality and quantity were measured by a Titertek-Berthold Colibri spectrometer (Colibri, Berlin, Germany). The absorbance ratios in 260/280 nm of RNA solution were from 1.9 to 2.0, which suggested high purity of the RNA samples. cDNA was synthesized using a reverse-transcribed kit with gDNase (Tiangen, Beijing, China). The primers of reference genes (*β-actin* and *rpl19*) and target genes in Table 2 were designed in NCBI. The reaction system of qPCR was 10 μL, including 1 μL cDNA (20 ng), 5 μL 2 × SYBR Mixture, 0.5 μL qPCR primers (10 μM), and 3.5 μL nuclease-free water. The qPCR program had 95 °C for 30 s, 40 cycles of 94 °C for 5 s, and 60 °C for 30 s, and was carried out in 96-well plates on a Roche LightCycler 96 system (Roche, Basel, Switzerland). The specificity of amplified products was detected by the melting curve at the end of the PCR. The amplification efficiency of primers was between 90% and 110%. The amplification efficiency was calculated using the following equation: E = 10 ^(−1/Slope)^ − 1 [33], and the target genes expression was calculated using the 2^−ΔΔCt^ method [34].

### 2.6. Fatty Acid Compositions Analysis

The liver was weighed and broken, and the total lipid was extracted using the chloroform–methanol (2:1, *v*/*v*) method. A total of 5 mg lipid was esterified with KOH–methanol and Boron trifluoride–methanol in 65 °C and 75 °C water baths for 30 min, respectively. Then, fatty acid methyl esters were extracted with hexane and assayed by gas chromatography (GC-2010 pro, Shimadzu, Kyoto, Japan) with a fused silica capillary column (SH-RT-2560, 100 m × 0.25 mm × 0.20 μm). The temperature programming of the column was listed as follows: 150 °C to 200 °C for 15 min and 200 °C to 250 °C for 2 min. Injector and detector temperatures were 250 °C. The contents of fatty acids were expressed as a percentage of each fatty acid with respect to the total fatty acids (%).

### 2.7. Statistical Analysis

All data are presented as means ± standard error of means (SEM) and were analyzed using the SPSS Statistics 21.0 software (IBM corporation, Armonk, NY, USA). All data were tested for normality and homogeneity of variances using Shapiro–Wilk and Levene’s tests. One-way analysis of variance (ANOVA) was performed to determine significant differences among the three treatments, followed by a Duncan’s multiple comparison test. Significant differences were set at *p* < 0.05.

## 3. Results

### 3.1. Effect of Intermittent and Chronic Hypoxia on Fish Size and Organ Weight

Compared with the normoxia group (NO), the final body weight, weight gain, and viscerasomatic index of the fish in the intermittent hypoxia group (IH) and chronic hypoxia group (CH) were decreased significantly (Figure 1A,B,D). Meanwhile, the feed conversion ratio in the IH and CH groups was significantly higher than in the NO group (Figure 1C). Compared with the NO group, the hepatosomatic index and condition factor were reduced in the IH and CH groups, and significance was found in the hepatosomatic index of the CH group (Figure 1E,F). IH and CH (DO = 3.5 ± 0.5 mg/L) did not affect the survival of the tiger puffer during the 4-week farming (Figure 1G). These data proved that intermittent and chronic hypoxia conditions decreased fish growth and visceral weight.

### 3.2. Effects of Intermittent and Chronic Hypoxia on Whole Fish Compositions

Compared with the NO group, the whole-fish moisture and crude lipid contents in the IH and CH groups showed an increased trend and a decreased trend, respectively (Figure 2A,B). The crude protein content of whole fish in the CH group was significantly higher than the NO and IH groups (Figure 2C). These results suggest that chronic hypoxia decreased the crude lipid content and increased the crude protein of whole fish.

### 3.3. Effects of Intermittent and Chronic Hypoxia on Glucose Metabolism

The CH group had significantly higher contents of glucose and lactate contents in the serum, as well as higher lactate content in the liver than the NO group, but these indices were not significantly different between the IH and NO groups (Figure 3A,C,F). In addition, IH and CH had no significant effects on the pyruvate, glycogen, and lactate contents in the serum liver or muscle (Figure 3B,D,E,G,H). These data indicate that chronic hypoxia, but not intermittent hypoxia, promoted the conversion of glucose to lactate.

### 3.4. Effects of Intermittent and Chronic Hypoxia on Hifα and Glycolysis Pathway

Compared with the NO and IH groups, the gene expressions of hypoxia-inducible factor 1 subunit alpha a (*hif1α*), *hif2α*, glucokinase (gck), phosphofructokinase (pfk), lactate dehydrogenase A4 (ldha), and gsk3β in the liver, as well as hif3α in the muscle, were upregulated significantly in the CH group (Figure 4A,B,E–H). However, there was no significant difference in these indicators between the NO and IH groups. The IH and CH groups all had significantly lower *hif1α*, *hk1*, and *gsk3β* gene expressions in the muscle than the NO group (Figure 4I,M,O). The expression of the *vegfa* gene in the liver was significantly lower in the IH group than that in the NO group (Figure 4D). In addition, IH and CH did not affect the expression of *hif3α* in the liver and *hif2α*, *vegfa*, and *ldha* in the muscle (Figure 4C,J,L,N). All these results indicate that chronic hypoxia promoted the expression of *hif1α*, *hif2α*, and glycolysis-related genes in the liver.

### 3.5. Effects of Intermittent and Chronic Hypoxia on Lipid Metabolism

Compared with the NO group, both IH and CH decreased the crude lipid and triglyceride levels in the liver, and a significant difference was found in the crude lipid content of the IH group (Figure 5B,C). However, there was no significant difference in the triglyceride level of serum and muscle between NO and hypoxia groups (Figure 5A,D). Compared with the NO group, the expression of lipid synthesis-related genes (*fasn* and *srebf1*) in the liver showed a reducing trend in the IH group but was upregulated in the CH group (Figure 5E,F). Both IH and CH did not affect the expression of lipid catabolism-related genes (*atgl* and *cpt1ab*) in the liver and lipid synthesis-related genes (*fasn* and *srebf1*) in the muscle (Figure 5G–J). Compared with the NO group, the expression of lipid catabolism-related genes (*atgl* and *cpt1b*) in the muscle in the IH and CH groups were downregulated significantly (Figure 5K,L). These data suggest that intermittent hypoxia and chronic hypoxia can reduce the lipid content of tiger puffer.

### 3.6. Effects of Intermittent and Chronic Hypoxia on Fatty Acids Compositions

Compared with the NO group, the C16:0 level was reduced in the IH and CH groups (Table 3). IH decreased the monounsaturated fatty acid (MUFA) content but increased the contents of n-3 polyunsaturated fatty acid (n-3PUFA), including C20:5n-3 (EPA), C22:5n-3 (DPA), and C22:6n-3 (DHA). IH and CH did not affect the levels of saturated fatty acid (SFA) and n-6 polyunsaturated fatty acid (n-6PUFA).

### 3.7. Effects of Intermittent and Chronic Hypoxia on Protein Metabolism

The total soluble protein level in the serum and liver, as well as blood hemoglobin content in the IH and CH groups, was higher than that in the NO group (Figure 6A–C). However, the total soluble protein content in the muscle was not affected by IH and CH (Figure 6D). Compared with the NO and IH groups, the expression of *ir*, *akt1*, mechanistic target of rapamycin kinase (*mtor*), and ubiquitin-like modifier activating enzyme 1 (*uba1*) in the liver of the CH group were upregulated, and significant differences were found in *mtor* and *uba1* (Figure 6E–H). However, the expression of *akt1* and *uba1* in the muscle of both IH and CH groups was lower than that in the NO group, and a significant difference was found in *uba1* (Figure 6J,L). In addition, there were no significant differences in *ir* and *mtor* expression in the muscle between the NO and hypoxia groups (Figure 6I,K). All these data indicate that intermittent and chronic hypoxia could promote the synthesis of protein in the liver.

## 4. Discussion

In natural environments and aquaculture activities, hypoxia (oxygen deficiency) widely exists and is harmful to aquatic animals. Since the middle of the 20th century, eutrophication and global warming caused by excess nutrient loading and fossil fuel burning generated more chronic hypoxia and anoxia environments in shallow coastal and estuarine areas [35,36]. The northern Gulf of Mexico, as well as the Black Sea and Baltic Sea, are the typical ecosystems that were burdened with severe seasonal hypoxia, leading to alteration of food webs and loss of fisheries and biodiversity [37]. In Snug Harbor (MA, USA) and Sanya Bay (Hainan, China), DO and pH changes both showed a distinct diurnal cycle caused by photosynthesis of benthic microalgae and macroalgae, with DO and pH rising at daytime and falling at night [7,8]. A similar phenomenon was also found in freshwater ponds and lakes [6]. So many aquatic animals live in intermittent hypoxia environments, and the physiological strategies for adapting to intermittent hypoxia of tiger puffer are poorly understood. In addition, the different responses of fish to intermittent hypoxia and chronic hypoxia are not clear.

Previous studies have shown that in intermittent hypoxia, the oxygen consumption rate of estuarine killifish (*Fundulus heteroclitus*) significantly decreased in the nighttime-hypoxia phase with 0.8 mg/L O_2_, then increased in the daytime-reoxygenation phase with 8 mg/L O_2_, but the oxygen consumption rate was consistently at a low level in the chronic hypoxia condition [38]. For largemouth bass, chronic hypoxia (17–26% DO saturation) for 6 weeks significantly decreased the weight gain, specific growth rate, and feed intake [39]. In European sea bass (*Dicentrarchus labrax*), chronic moderate hypoxia (40% DO saturation) for 38 days reduced the fish size and protein digestibility by decreasing the trypsin and amylase activities in the liver, as well as alkaline phosphatase and aminopeptidase-N activities in the intestine [40]. Similar results were also found in zebrafish (*Danio rerio*) [33] and oriental river prawn (*Macrobrachium nipponense*) [41]. In Nile tilapia, compared with normoxia group (*p*O_2_ = 17.4 ± 0.4 kPa), chronic hypoxia (*p*O_2_ = 8.1 ± 0.6 kPa) for 9 weeks reduced the feed intake, apparent digestibility of nutrients, weight gain, hepatosomatic index and viscerosomatic index, but intermittent hypoxia (6 h of hypoxia every night, *p*O_2_ = 0.4 ± 1.0 kPa) significantly promoted the growth and weight gain by increasing feed intake compensatorily [42]. In juvenile qingbo (*Spinibarbus sinensis*), intermittent hypoxia (DO = 7.0 mg/L from 7:00 to 21:00; DO = 3.0 mg/L from 21:00 to 7:00) for 30 days also significantly increased the weight gain and specific growth rate [43]. However, in the present study, both the intermittent hypoxia group and chronic hypoxia group reduced the final mean body weight, weight gain, viscerasomatic index, hepatosomatic index, and condition factor of tiger puffer compared to the normoxia group. Meanwhile, compared with the normoxia condition, chronic hypoxia for 4 weeks had more significant negative effects on fish growth and visceral weight than intermittent hypoxia. The possible reason could be that chronic hypoxia leads to lower oxygen consumption and metabolic rate. Intermittent hypoxia (44–65% DO saturation, average 4 h every day) for 38 days also decreased the feed intake and weight gain in Atlantic salmon (*Salmo salar*) [44]. The possible reasons for the different growth performances of different fish species in intermittent hypoxia could be that Nile tilapia and qingbo fish are very tolerant to hypoxia, but tiger puffer and Atlantic salmon are intolerant and sensitive to hypoxia. Nile tilapia and qingbo fish can recover from intermittent hypoxia quickly, using the saved basal energy expenditure and active metabolic expenditure for growth, and increase feed intake compensatorily from intermittent hypoxia to normoxia condition.

Non-cyprinidae fish species and mammals have three *hifα* isoforms, including *hif1α*, *hif2α*, and *hif3α*. The partial functions of these isoforms have been identified in several fish species, such as zebrafish [45], Nile tilapia [46], blunt snout bream (*Megalobrama amblycephala*) [47], turbot [48], estuarine fish (*Fundulus heteroclitus*) [49], Amur minnow (*Phoxinus lagowskii*) [50], and thick-lipped grey mullet (*Chelon labrosus*) [51]. However, the functions of the three *hifα* isoforms of tiger puffer in intermittent hypoxia and chronic hypoxia are still unclear. Studies have found that the *hif1α* and *hif2α* expression in the liver was increased several-fold in dragonet (*Callionymus valenciennei*) that lived in hypoxic areas of Tokyo Bay, and the *hif1α* and *hif2α* expressions were decreased after reoxygenation for 24 h [52]. In Korean rockfish (*Sebastes schlegeli*), acute hypoxia for 30–60 min significantly increased the *hif1α* mRNA expression in the ovary and the *hif2α* mRNA expression in the gill and spleen but decreased the *hif2α* mRNA expression in the gonads and liver [53]. In C57BL/6 male mice and human HepG2 cells, inhibition of *hif2α* reversed the liver fat accumulation induced by hypoxia, which indicates that *hif2α* plays an important role in lipid synthesis under hypoxia conditions [54]. Our previous studies in Nile tilapia and turbot have shown that the hepatic expression of *hif3α* but not *hif1α* was significantly upregulated under acute hypoxia, which suggests the *hif3α* is a marker gene for fish response to acute hypoxia [46,48]. In the present study, the CH group, but not the IH group, had significantly higher *hif1α*, *hif2α*, *gck*, *pfk*, *ldha*, and *gsk3β* expression in the liver than the NO group. However, the IH and CH groups had significantly lower *hif1α*, *hk1*, and *gsk3β* expression in the muscle than the NO group, and IH and CH did not affect the *hif3α* expression in the liver. These results showed that chronic hypoxia had more significant effects than intermittent hypoxia on fish, and the liver is more sensitive to hypoxia than the muscle. Moreover, *hif1α* and *hif2α* play key roles in the adaptation of tiger puffer to chronic hypoxia.

Many studies have described the changes in nutrient metabolism under chronic hypoxia or intermittent hypoxia, but the differences in nutrient metabolism between intermittent hypoxia and chronic hypoxia have not been known in marine fish species. In largemouth bass, chronic hypoxia (17–26% DO saturation) for 6 weeks significantly decreased the myofiber diameter and protein synthesis by inhibiting the HIF1α/REDD1/mTOR pathway and increased the protein degradation and SOD, GPX, and CAT enzymes activities in the serum and muscle [39]. In juvenile turbot and European sea bass, chronic hypoxia (DO = 3.2–4.5 mg/L) for 42 days reduced the liver glycogen content and increased the serum lactate and blood hematocrit levels but did not change the serum glucose content [55]. In killifish (*Fundulus heteroclitus*), chronic hypoxia (*p*O_2_ = 5 kPa for 28 days) but not intermittent hypoxia (*p*O_2_ = 5 kPa at night, *p*O_2_ = 20 kPa at day for 28 days) reduced the total length of gill filaments and the number of oxidative fibers of the swimming muscle and increased the blood hemoglobin level [56]. However, intermittent hypoxia, not constant hypoxia, increased the capillary density in the glycolytic muscle, the protein content, and the cytochrome c oxidase activity in the liver [56]. In high-latitude fish (*Phoxinus lagowskii*), the metabolomics showed that compared with normoxia, both chronic hypoxia (DO = 3.0–4.0 mg/L for 28 days) and intermittent hypoxia (DO = 6.0–7.0 mg/L from 07:00 to 21:00, DO = 3.0–4.0 mg/L from 21:00 to 07:00 for 28 days) changed the starch and sucrose metabolism, carbohydrate digestion and absorption, and ATP binding cassette transporters pathways, but the significantly changed pathways between the chronic hypoxia and intermittent hypoxia were biosynthesis of secondary metabolites, ATP binding cassette transporters, and bile secretion [57]. In *Phoxinus lagowskii*, intermittent hypoxia increased the TG level, PK and LDH activities, *hif1α*, *hif2α*, *hif3α*, *vhl*, *fih* expression in the heart, as well as the MDA level and the SOD, CAT, LDH activities in the brain [58]. However, chronic hypoxia (DO = 3–4 mg/L) increased the *hif1α*, *hif2α*, *hif3α*, *vhl*, and *fih* expression in the brain and did not change the oxidative stress biomarkers in the brain [58]. Some different results were found in the present study. Chronic hypoxia increased the blood hemoglobin content, the total soluble protein level in the serum and liver, *mtor* (protein synthesis-related) expression, and crude protein of whole fish. Meanwhile, chronic hypoxia increased the lactate content in the serum and liver and promoted the conversion of glucose to lactate. However, intermittent hypoxia significantly reduced the crude lipid and triglyceride contents, expression of lipid synthesis-related genes (*fasn* and *srebf1*) in the liver, and crude lipid content of tiger puffer fish but elevated the n-3 polyunsaturated fatty acid (PUFA) proportion and reduced the monounsaturated fatty acid (MUFA) proportion. Therefore, fish species, stress time and degree, and tissue difference all affect the physiological and metabolic responses of fish to hypoxia.

## 5. Conclusions

Intermittent and chronic hypoxia are common and harmful to marine animals. The present study suggested that both intermittent and chronic hypoxia decreased fish growth and visceral weight but increased the feed conversion ratio and blood hemoglobin content. Chronic hypoxia but not intermittent hypoxia promoted the protein synthesis by activating *mtor* expression and promoted the glycolysis pathway by activating *hif1α* and *hif2α* expression. Intermittent hypoxia but not chronic hypoxia decreased the lipid synthesis by inhibiting *fasn* and *srebf1* expression. Meanwhile, intermittent hypoxia reduced the monounsaturated fatty acid content but increased the contents of n-3 polyunsaturated fatty acid (EPA, DPA, and DHA). These changes promoted the adaption of tiger puffer to intermittent and chronic hypoxia.

## Figures and Tables

**Figure 1 animals-14-02470-f001:**
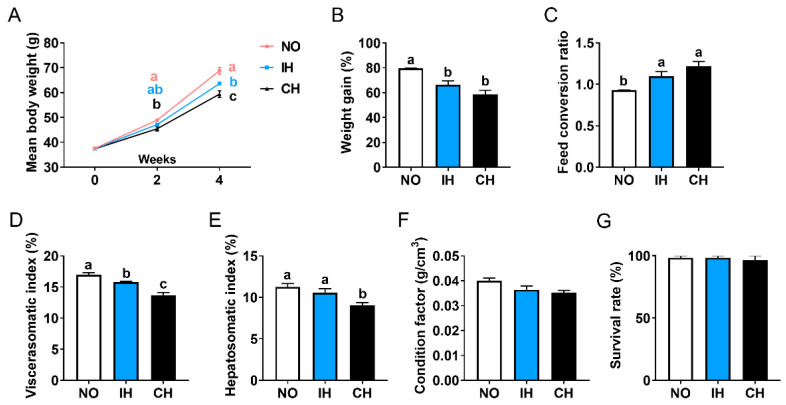
Effect of intermittent and chronic hypoxia on growth and visceral weight in tiger puffer (*Takifugu rubripes*). (**A**) Final body weight; (**B**) Weight gain; (**C**) Feed conversion ratio; (**D**) Viscerasomatic index; (**E**) Hepatosomatic index; (**F**) Condition factor; (**G**) Survival. Note: NO: normoxia group; IH: intermittent hypoxia group; CH: chronic hypoxia group. Different letters above the bars suggest significant differences (*p* < 0.05) among the three groups (mean ± SEM, n = 3–6).

**Figure 2 animals-14-02470-f002:**
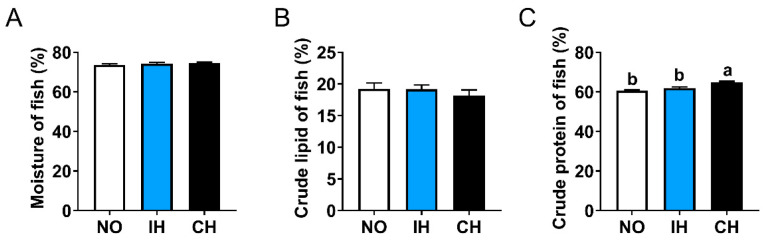
Effects of intermittent and chronic hypoxia on whole-fish proximate compositions in tiger puffer (*Takifugu rubripes*). (**A**) Moisture; (**B**) Crude lipid; (**C**) Crude protein. Note: NO: normoxia group; IH: intermittent hypoxia group; CH: chronic hypoxia group. Different letters above the bars suggest significant differences (*p* < 0.05) among the three groups (mean ± SEM, n = 6).

**Figure 3 animals-14-02470-f003:**
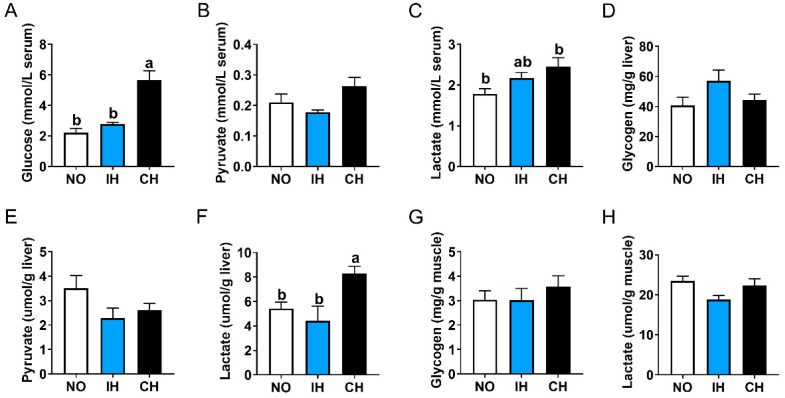
Effects of intermittent and chronic hypoxia on glucose metabolism in tiger puffer (*Takifugu rubripes*). (**A**) Serum glucose content; (**B**) Serum pyruvate content; (**C**) Serum lactate content; (**D**) Liver glycogen content; (**E**) Liver pyruvate content; (**F**) Liver lactate content; (**G**) Muscle glycogen content; (**H**) Muscle lactate content. Note: NO: normoxia group; IH: intermittent hypoxia group; CH: chronic hypoxia group. Different letters above the bars suggest significant differences (*p* < 0.05) among the three groups (mean ± SEM, n = 6).

**Figure 4 animals-14-02470-f004:**
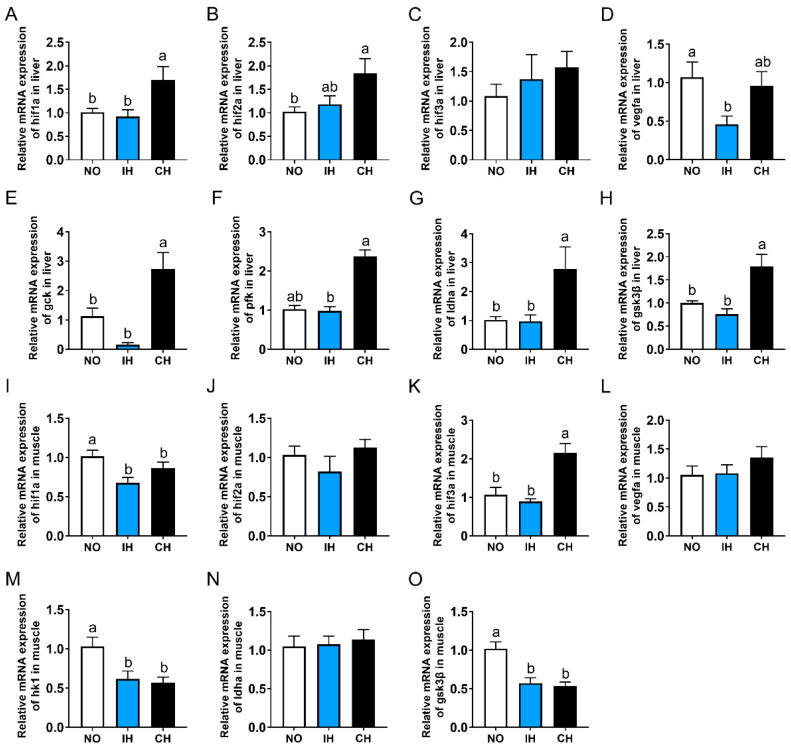
Effects of intermittent and chronic hypoxia on *hifα* and glycolysis pathway in tiger puffer (*Takifugu rubripes*). (**A**–**D**) Liver *hif1α*, *hif2α*, *hif3α* and *vegfa* expression; (**E**–**H**) Liver *gck*, *pfk*, *ldha* and *gsk3β* expression; (**I**–**L**) Muscle *hif1α*, *hif2α*, *hif3α* and *vegfa* expression; (**M**–**O**) Muscle *hk1*, *ldha* and *gsk3β* expression. Note: NO: normoxia group; IH: intermittent hypoxia group; CH: chronic hypoxia group. Different letters above the bars suggest significant differences (*p* < 0.05) among the three groups (mean ± SEM, n = 6).

**Figure 5 animals-14-02470-f005:**
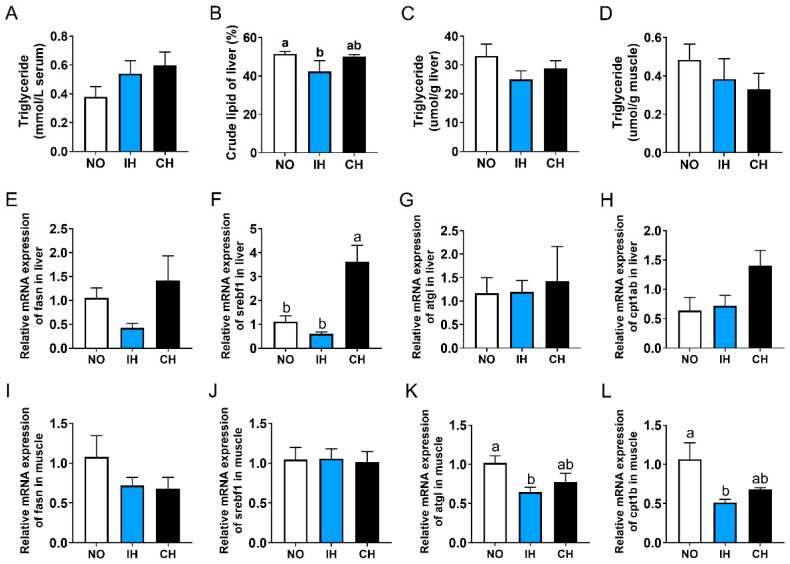
Effects of intermittent and chronic hypoxia on lipid metabolism in tiger puffer (*Takifugu rubripes*). (**A**) Serum triglyceride content; (**B**) Liver crude lipid content; (**C**) Liver triglyceride content; (**D**) Muscle triglyceride content; (**E**) Liver *fasn* expression; (**F**) Liver *srebf1* expression; (**G**) Liver *atgl* expression; (**H**) Liver *cpt1ab* expression; (**I**) Muscle *fasn* expression; (**J**) Muscle *srebf1* expression; (**K**) Muscle *atgl* expression; (**L**) Muscle *cpt1b* expression. Note: NO: normoxia group; IH: intermittent hypoxia group; CH: chronic hypoxia group. Different letters above the bars suggest significant differences (*p* < 0.05) among the three groups (mean ± SEM, n = 6).

**Figure 6 animals-14-02470-f006:**
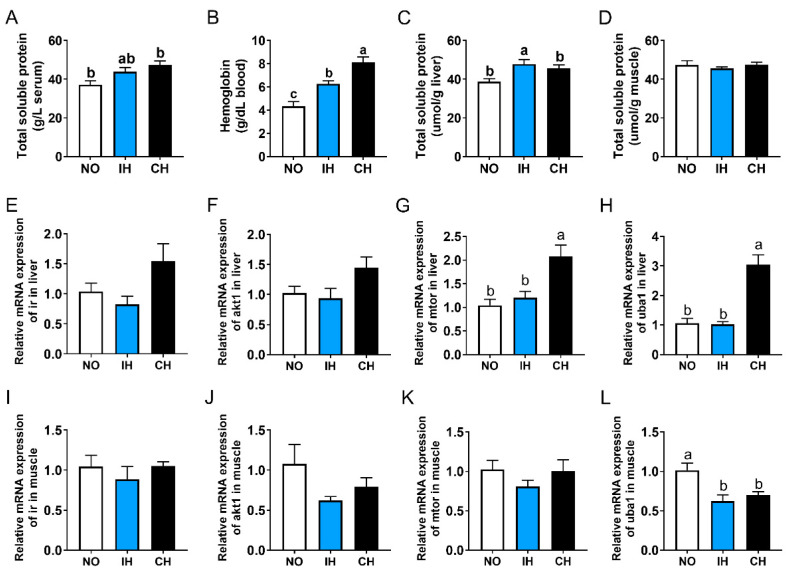
Effects of intermittent and chronic hypoxia on protein metabolism in tiger puffer (*Takifugu rubripes*). (**A**) Serum total soluble protein content; (**B**) Blood hemoglobin content; (**C**) Liver total soluble protein content; (**D**) Muscle total soluble protein content; (**E**–**H**) Liver *ir*, *akt1*, *mtor*, and *uba1* genes expression; (**I**–**L**) Muscle *ir*, *akt1*, *mtor*, and *uba1* genes expression. Note: NO: normoxia group; IH: intermittent hypoxia group; CH: chronic hypoxia group. Different letters above the bars suggest significant differences (*p* < 0.05) among the three groups (mean ± SEM, n = 6).

**Table 1 animals-14-02470-t001:** Formula and proximate compositions of the experimental diets used for tiger puffer (*Takifugu rubripes*).

Dietary Ingredient (% of Dry Matter)	
Fishmeal	41
Soybean protein concentrate	25
Wheat meal	21
Fish oil	6
Soybean lecithin	1.5
Vitamin premix ^a^	0.4
Mineral premix ^b^	0.8
Choline chloride	0.5
Butylated hydroxytoluene	0.02
Dimethyl-beta-propiothetin	0.1
Ca(H_2_PO_4_)_2_	1.5
Vitamin C	0.5
Carboxymethyl cellulose	1.68
Total	100
Proximate compositions:	
Moisture (%)	9.31
Crude fat (%)	10.62
Crude protein (%)	49.46
Ash (%)	8.65

^a^ Vitamin premix and ^b^ mineral premix, designed for marine fish, were purchased from Qingdao Master Biotech Co., Ltd., Qingdao, China.

**Table 2 animals-14-02470-t002:** Primers used for the analysis of gene expression for tiger puffer (*Takifugu rubripes*).

Gene Name	Sequences (5′ to 3′) Forward and Reverse	Product Length	GenBank NO.
*hif3α/hif1αl*	AAGCATCAGCATCAAACGGAG	111	XM_011608719.2
(hypoxia-inducible factor 1-alpha-like)	GTGTGGGCGAGCTCATAAAAC		
*hif2α/epas1b*	CACATGTGCAGAATCCCCCT	191	XM_011603052.2
(endothelial PAS domain-containing protein 1)	ATGGGGTATGCTCTGTTGGC		
*hif1* *α*	CCCCCTTCAGCTTACCAGAC	157	XM_003962474.3
(hypoxia-inducible factor 1 subunit alpha)	CTTTTGCGTGGGGTGTTCTG		
*vegfa*	CATATCACGATGCCGTTTGTG	162	XM_029849217.1
(vascular endothelial growth factor A)	CACATTTGCAGGTCCGTTCG		
*gck*	GAGGACTGTGGAACTGGTGG	81	XM_029829322.1
(glucokinase)	TCTCCATTGTCCCCGAATGC		
*hk1*	GGTTGAGGACCACCATAGGC	100	XM_003969460.3
(hexokinase 1)	ATATCCGGGACCAAACGACG		
*pfk*	GAATGGGCATCTACGTGGGG	140	XM_029844357.1
(ATP-dependent 6-phosphofructokinase, liver type)	ACCTATCATGGTCCCACCCT		
ldha	GCGTCACCGCTAATTCCAAG	193	XM_003967364.3
(L-lactate dehydrogenase A chain)	CAGGCCACGTAGGTCAGAAT		
* srebf1*	CGAGTGTGGAGCAGCCTAAA	170	XM_011603881.2
(sterol regulatory element binding transcription factor 1)	AGGGCTCTGGGTCTGAATCT		
*fasn*	GGAGCTGACTACAAGCTGGG	81	XM_011619859.2
(fatty acid synthase)	CAGGAAGGTTCGGTGGTCTC		
*cpt1ab*	CCTGATGGATGAAGAGCGGT	111	XM_011607269.2
(carnitine O-palmitoyltransferase 1, liver isoform-like)	GAGGCCCACCAGGATTTGAG		
*cpt1b*	TCTATCCCGCCAGTCCATCT	115	XM_029841851.1
(carnitine O-palmitoyltransferase 1, muscle isoform)	GGCAGGTTCTCCTTCATTGC		
*atgl/pnpla2*	CGCCGTGGAACATTTCGTTT	84	XM_003967696.3
(patatin-like phospholipase domain containing 2)	GCGCTTGTTCCAACAGACAG		
* uba1*	TTTCATTGGCGGTTTGGCTG	156	XM_029834111.1
(ubiquitin-like modifier activating enzyme 1)	CGGCAGTTTCTAGGAGCACA		
*mtor*	CGCCTTCCTCTCTTGTTGGT	162	XM_011621515.2
(mechanistic target of rapamycin kinase)	AGGGGTAGAGGACCCTTGTC		
*ir*	AGCAAGGACATCCGGAACAG	113	XM_003975383.3
(insulin receptor)	CGGAAATCCTCTGGCTTGGT		
* akt1*	GGAGACGGACACGCGATATT	139	XM_029832808.1
(AKT serine/threonine kinase 1)	ACTGGCGGAGTAGGAGAACT		
*gsk3β*	ATCAAGGTTCTGGGCACACC	126	XM_029839479.1
(glycogen synthase kinase-3 beta)	TGGTCGGAATACCTGCTGAC		
* rpl19*	GATCCCAACGAGACCAACGA	191	XM_003964816.3
(60S ribosomal protein L19)	CGAGCATTGGCTGTACCCTT		
* β-actin*	GGAAGATGAAATCGCCGCAC	196	XM_003964421.3
(actin beta)	GGTCAGGATACCCCTCTTGC		

**Table 3 animals-14-02470-t003:** Liver fatty acid compositions of tiger puffer (*Takifugu rubripes*) (% total fatty acids).

Fatty Acid	NO	IH	CH
C14:0	3.68 ± 0.04	3.62 ± 0.09	3.63 ± 0.08
C16:0	20.48 ± 0.32	19.34 ± 0.24	19.6 ± 0.46
C17:0	0.40 ± 0.01	0.4 ± 0.02	0.41 ± 0.02
C18:0	7.47 ± 0.20	7.36 ± 0.06	7.39 ± 0.20
C20:0	0.26 ± 0.02	0.24 ± 0.01	0.26 ± 0.01
∑SFA	32.29 ± 0.32	30.95 ± 0.29	31.77 ± 0.30
C14:1n-5	0.43 ± 0.01	0.42 ± 0.02	0.42 ± 0.01
C16:1n-7	7.89 ± 0.22	7.76 ± 0.07	7.75 ± 0.12
C17:1n-7	0.38 ± 0.01	0.42 ± 0.01	0.4 ± 0.01
C18:1n-9	24.71 ± 0.23	23.89 ± 0.32	24.66 ± 0.37
C20:1n-9	1.61 ± 0.06	1.70 ± 0.08	1.56 ± 0.05
C22:1n-9	0.16 ± 0.02	0.18 ± 0.01	0.20 ± 0.01
C24:1n-9	0.29 ± 0.04	0.26 ± 0.02	0.30 ± 0.02
∑MUFA	35.41 ± 0.13 ^a^	34.58 ± 0.27 ^b^	35.21 ± 0.24 ^ab^
C18:2n-6	11.04 ± 0.18	11.27 ± 0.29	11.11 ± 0.15
C18:3n-6	0.40 ± 0.02	0.39 ± 0.02	0.38 ± 0.03
C20:2n-6	0.62 ± 0.02	0.72 ± 0.03	0.71 ± 0.04
C20:3n-6	0.71 ± 0.04	0.65 ± 0.03	0.71 ± 0.04
C22:2n-6	0.43 ± 0.02	0.47 ± 0.02	0.39 ± 0.07
∑n-6PUFA	13.2 ± 0.16	13.5 ± 0.35	13.3 ± 0.26
C20:3n-3	0.58 ± 0.02 ^b^	0.55 ± 0.02 ^b^	0.68 ± 0.03 ^a^
C20:5n-3	5.50 ± 0.13	6.10 ± 0.13	5.95 ± 0.21
C22:5n-3	4.05 ± 0.16 ^b^	4.63 ± 0.09 ^a^	4.00 ± 0.17 ^b^
C22:6n-3	8.91 ± 0.12 ^b^	9.69 ± 0.09 ^a^	9.55 ± 0.24 ^ab^
∑n-3PUFA	19.03 ± 0.33 ^b^	20.97 ± 0.23 ^a^	20.17 ± 0.58 ^ab^

SFA: saturated fatty acid; MUFA, monounsaturated fatty acid; PUFA: polyunsaturated fatty acid. Different letters above the bars suggest significant differences (*p* < 0.05) among the three groups (mean ± SEM, n = 6).

## Data Availability

The datasets generated during the current study are available from the corresponding author upon reasonable request.

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
