# Peer review of "Effects of Intermittent and Chronic Hypoxia on Fish Size and Nutrient Metabolism in Tiger Puffer (Takifugu rubripes)"

_animals, 2024, doi:10.3390/ani14172470_

Round 1

Reviewer 1 Report

Comments and Suggestions for Authors

Below my comments to the MS Animals-3154360

In my opinion, this is an excellent MS. The titles of tables and figures require improvement and supplementation. They should be much more informative. They should include the English and Latin name of the studied fish species. The authors should take into account that tables and figures can "live their own life" and should be fully informative for the reader.

Hypoxia in marine fish, compared to freshwater fish, is relatively rarely studied. The reason is the issue of different adaptations to environmental conditions and the fact that in the freshwater environment oxygen deficiencies occur much more often than in the marine environment. Hence, the issue of studying hypoxia and chronic hypoxia and its effect on metabolism in marine fish is extremely important, also in terms of lowering the oxygen level in intensive marine fish aquaculture.

Additional detailed comments on the titles of Tables and Figures

Table 1. The title should be more informative, i.e. Formulation and approximate composition of the experimental diets used for ..... Please also add the English and Latin names of studied fish species.

Table. 2 - the same comments as for the title of Table 1. It should be remembered that tables and figures can "live their own life" and must be informative not only within the text of the publication, but also outside it.

Figure 1. Please add the English and Latin name of the studied fish species and add details, e.g. in different oxygen level conditions.

Figure 2; Figure 3; Figure 4; Figure 5; Figure 6. The same comments as for Figure 1.

Table 3. the same comments as for Table 1 (although the English name is already there). It is necessary to explain what it is (% TEA) and explain the letter designations (statistics) inside the table.

Author Response

Response to the comments of reviewers (point by point)

We sincerely thank reviewers to point out those detailed mistakes. All these mistakes have been corrected and the details are shown below. All changes are highlighted with yellow in the text.

In my opinion, this is an excellent MS. The titles of tables and figures require improvement and supplementation. They should be much more informative. They should include the English and Latin name of the studied fish species. The authors should take into account that tables and figures can "live their own life" and should be fully informative for the reader.

Additional detailed comments on the titles of Tables and Figures

  1. Table 1. The title should be more informative, i.e. Formulation and approximate composition of the experimental diets used for ..... Please also add the English and Latin names of studied fish species.

Response: Done, it is good suggestion. We have added the English and Latin names of studied fish species in table 1.

  1. Table. 2 - the same comments as for the title of Table 1. It should be remembered that tables and figures can "live their own life" and must be informative not only within the text of the publication, but also outside it.

Response: Done, we have added the English and Latin names of studied fish species in table 2.

  1. Figure 1. Please add the English and Latin name of the studied fish species and add details, e.g. in different oxygen level conditions.

Response: Done, we have added the English and Latin names of studied fish species in figure 1. Because the words “Effect of intermittent and chronic hypoxia on” are already included in the title of figure 1, there is no need to add “in different oxygen level conditions” again.

  1. Figure 2; Figure 3; Figure 4; Figure 5; Figure 6. The same comments as for Figure 1.

Response: Done, we have added the English and Latin names of studied fish species in figures 2-6.

  1. Table 3. the same comments as for Table 1 (although the English name is already there). It is necessary to explain what it is (% TEA) and explain the letter designations (statistics) inside the table.

Response: Done, we have added the Latin names of tiger puffer, explained what it is (% TFA) and the letter designations in table 3.

Reviewer 2 Report

Comments and Suggestions for Authors

This study evaluated the effects of intermittent and chronic hypoxia on fish size and nutrient metabolism of tiger puffer, which would improve our understanding of the adaptive mechanisms to different hypoxia patterns in fish. The manuscript is well organized and the conclusion is well supported. There are some detail suggestions and comments as follow.

 1.      In abstract, line 35-36, “which provide important information about manipulation of hypoxia tolerance in fish.” the sentence is difficult to understand. Please revised it.

2.      The dissolved oxygen level of hypoxia groups was 3.5 ± 0.5 mg/L. Is this dissolved oxygen level too high? Intermittent hypoxia group, how did authors regulate the dissolved oxygen levels at day and night?

3.      In results, “compared with the normoxia group (NO), the final body weight, weight gain and viscerasomatic index of the fish in the intermittent hypoxia group (IH) and chronic hypoxia group (CH) were decreased significantly.”; “The crude protein content of whole fish in the CH group was significantly higher than the NO and IH groups.”; “The total soluble protein level in the serum and liver, and blood hemoglobin con-tent in the IH and CH groups were higher than that in the NO group”. Why did the body weight decrease while the protein content increased?4.      In table1, tiger puffer is a carnivorous fish. Is the 21% wheat meal in the diet too high?

5.      In figure 5 and table3, compared with the NO group, IH group decreased the crude lipid and triglyceride levels in the liver, but increased the contents of n-3 polyunsaturated fatty acid (n-3PUFA), including C20:5n-3 (EPA), C22:5n-3 (DPA) and C22:6n-3 (DHA). How to explain this phenomenon?

6.      Line 66, “produced” should be “produce”.

7.      Line 140, “that made following” should be “followed”.

8.      In 2.4. Biochemical indexes assays, please state the product codes of the commercial kits.

9.      Line 211, “were” should be “was”.

10.  Line 259, add “the” before “three groups”.

11.  Line 354, “causing” should be “caused”.

12.  Line 362, “environment of intermittent hypoxia” should be “intermittent hypoxia environment”.

13.  Delete “(Micropterus salmoides)” in line 370, “(Danio rerio)” in line 400, “(Scophthalmus maximus)” in line 401, “(Dicentrarchus labrax)” in line 430. These Latin names were repeated in the previous text.

14. Line 425, “has” should be “have”. Delete “clearly”.

Comments on the Quality of English Language

Moderate editing of English language required.

Author Response

Response to the comments of reviewers (point by point)

We sincerely thank reviewers to point out those detailed mistakes. All these mistakes have been corrected and the details are shown below. All changes are highlighted with yellow in the text.

This study evaluated the effects of intermittent and chronic hypoxia on fish size and nutrient metabolism of tiger puffer, which would improve our understanding of the adaptive mechanisms to different hypoxia patterns in fish. The manuscript is well organized and the conclusion is well supported. There are some detail suggestions and comments as follow.

  1. In abstract, line 35-36, “which provide important information about manipulation of hypoxia tolerance in fish.” the sentence is difficult to understand. Please revised it.

Response: Done, we have revised the sentence to “which provide important information about mechanisms of hypoxia adaption in fish.”.

  1. The dissolved oxygen level of hypoxia groups was 3.5 ± 0.5 mg/L. Is this dissolved oxygen level too high? Intermittent hypoxia group, how did authors regulate the dissolved oxygen levels at day and night?

Response: Some omnivorous fish species, such as Carassius auratus and Oreochromis niloticus, can tolerate lower dissolve oxygen level. However, tiger puffer has only a slit-like gill opening anterior to the base of the pectoral fin, and is intolerance to hypoxia stress. Dissolve oxygen at 3-4 mg/L level is common in some aquaculture farms, and lower dissolve oxygen level (< 3 mg/L) for long time would cause severe growth inhibition and even massive dead of tiger puffer.

  1. In results, “compared with the normoxia group (NO), the final body weight, weight gain and viscerasomatic index of the fish in the intermittent hypoxia group (IH) and chronic hypoxia group (CH) were decreased significantly.”; “The crude protein content of whole fish in the CH group was significantly higher than the NO and IH groups.”; “The total soluble protein level in the serum and liver, and blood hemoglobin con-tent in the IH and CH groups were higher than that in the NO group”. Why did the body weight decrease while the protein content increased?

Response: This is an interesting phenomenon. Actually, in both fish and mammals, smaller or younger individuals tend to have higher body protein content. The main reason for this is that the body fat content of smaller or younger tiger puffer is lower.

  1. In table1, tiger puffer is a carnivorous fish. Is the 21% wheat meal in the diet too high?

Response: There was a reference for carbohydrate utilization of tiger puffer. Study found that appropriate supplementation of carbohydrate in diet promoted growth performance and feed utilization of juvenile tiger puffer. Based on the quadratic regression between SGR and carbohydrate levels, the optimal dietary carbohydrate level of juvenile tiger puffer is 21.6%. Therefore, we added 21% wheat meal in the experimental diet.

Reference:

Qingqing Guo, Yuetao Wang, Ning Li, Tao Li, Yujing Guan, Yonghui Wang, Peiyu Zhang, Zhi Li, Haiyan Liu. Effects of dietary carbohydrate levels on growth performance, feed utilization, liver histology and intestinal microflora of juvenile tiger puffer, Takifugu rubripes. Aquaculture Reports, 2024, 36: 102035. https://doi.org/10.1016/j.aqrep.2024.102035.

  1. In figure 5 and table3, compared with the NO group, IH group decreased the crude lipid and triglyceride levels in the liver, but increased the contents of n-3 polyunsaturated fatty acid (n-3PUFA), including C20:5n-3 (EPA), C22:5n-3 (DPA) and C22:6n-3 (DHA). How to explain this phenomenon?

Response: Study found that under hypoxia condition, saturated fatty acids would translate into toxic ceramides and acyl-carnitines. Triglycerides could counter a toxic buildup of saturated lipids, primarily by releasing the unsaturated fatty acid from lipid droplets into phospholipid pools. More EPA and DHA have stronger anti-inflammatory and antioxidant effects, which could promote tolerance and survival of cell under hypoxia.

Reference:

Daniel Ackerman, Sergey Tumanov, Bo Qiu et al. Triglycerides promote lipid homeostasis during hypoxic stress by balancing fatty acid saturation. 2018, Cell Reports, 24: 2596–2605. https://doi.org/10.1016/j.celrep.2018.08.015

  1. Line 66, “produced” should be “produce”.

Response: Done.

  1. Line 140, “that made following” should be “followed”.

Response: Done.

  1. In 2.4. Biochemical indexes assays, please state the product codes of the commercial kits.

Response: Done. We have added it.

  1. Line 211, “were” should be “was”.

Response: Done.

  1. Line 259, add “the” before “three groups”.

Response: Done.

  1. Line 354, “causing” should be “caused”.

Response: Done.

  1. Line 362, “environment of intermittent hypoxia” should be “intermittent hypoxia environment”.

Response: Done.

  1. Delete “(Micropterus salmoides)” in line 370, “(Danio rerio)” in line 400, “(Scophthalmus maximus)” in line 401, “(Dicentrarchus labrax)” in line 430. These Latin names were repeated in the previous text.

Response: Done. We have deleted it.

  1. Line 425, “has” should be “have”. Delete “clearly”.

Response: Done.